# Decentralized Training of Foundation Models in Heterogeneous Environments

**Binhang Yuan**[1]*, **Yongjun He**[1]*, **Jared Quincy Davis**[2], **Tianyi Zhang**[2], **Tri Dao**[2],
**Beidi Chen**[34], **Percy Liang**[2], **Christopher Re**[2], **Ce Zhang**[1]

[1]ETH Zürich, Switzerland   [2]Stanford University, USA   [3]Carnegie Mellon University   [4]Meta AI

{binhang.yuan, yongjun.he, ce.zhang}@inf.ethz.ch
{jaredq, tz58, trid, beidic, pliang, chrismre}@stanford.edu

## Abstract

Training foundation models, such as GPT-3 and PaLM, can be extremely expensive, often involving tens of thousands of GPUs running continuously for months. These models are typically trained in specialized clusters featuring fast, homogeneous interconnects and using carefully designed software systems that support both data parallelism and model/pipeline parallelism. Such dedicated clusters can be costly and difficult to obtain. *Can we instead leverage the much greater amount of decentralized, heterogeneous, and lower-bandwidth interconnected compute?* Previous works examining the heterogeneous, decentralized setting focus on relatively small models that can be trained in a purely data parallel manner. State-of-the-art schemes for model parallel foundation model training, such as Megatron and Deepspeed, only consider the homogeneous data center setting. In this paper, we present the first study of training large foundation models with model parallelism in a decentralized regime over a heterogeneous network. Our key technical contribution is a scheduling algorithm that allocates different computational "tasklets" in the training of foundation models to a group of decentralized GPU devices connected by a slow heterogeneous network. We provide a formal cost model and further propose an efficient evolutionary algorithm to find the optimal allocation strategy. We conduct extensive experiments that represent different scenarios for learning over geo-distributed devices simulated using real-world network measurements. In the most extreme case, across 8 different cities spanning 3 continents, our approach is $4.8\times$ faster than prior state-of-the-art training systems.

## 1   Introduction

Recent years have witnessed the rapid development of deep learning models, particularly foundation models (FMs) [1] such as GPT-3 [2] and PaLM [3]. Along with these rapid advancements, however, comes computational challenges in training these models: the training of these FMs can be very expensive — a single GPT3-175B training run takes 3.6K Petaflops-days [2]— this amounts to \$4M on today's AWS on demand instances, even assuming 50% device utilization (V100 GPUs peak at 125 TeraFLOPS)! Even the smaller scale language models, e.g., GPT3-1.3B (1.3 billion parameters), on which this paper evaluates, require 64 Tesla V100 GPUs to run for one week, costing \$32K on AWS. As a result, speeding up training and decreasing the cost of FMs have been active research areas. Due to their vast number of model parameters, state-of-the-art systems (e.g., Megatron[4], Deepspeed[5], Fairscale[6]) leverage multiple forms of parallelism [4, 7, 8, 9, 10, 11]. However, their design is only tailored to *fast*, *homogeneous* data center networks.

---

*Equal contribution.

36th Conference on Neural Information Processing Systems (NeurIPS 2022).

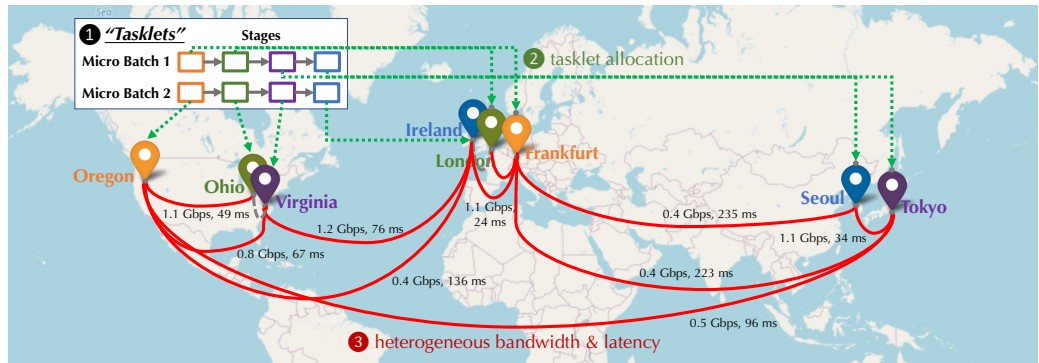

Figure 1: Given ① a set of computation tasklets involved in training foundation models (corresponding to different micro-batches and layers), and ② a heterogeneous network between devices, the goal is to find the optimal ③ allocation of tasklets to devices.

On the other hand, *decentralization* is a natural and promising direction. Jon Peddie Research reports that the PC and AIB GPU market shipped 101 million units in Q4 2021 alone [12]. Furthermore, many of these GPUs are underutilized. Leveraging this fact, volunteer computing projects such as Folding@Home [13] have sourced upwards of 40K Nvidia and AMD GPUs continuously [14]. Moreover, the incremental electricity and HVAC costs of running a V100 GPU for a volunteer are 50–100× lower than the spot prices for an equivalent device on AWS [15]. If we could make use of these devices in a decentralized open-volunteering paradigm for foundation model training, this would be a revolutionary alternative to the expensive solutions offered by data centers.

This vision inspired many recent efforts in decentralized learning, including both those that are theoretical and algorithmic [16, 17, 18], as well as recent prototypes such as Learning@Home [19] and DeDLOC [20]. However, efforts to-date in decentralized training either focus solely on *data parallelism* [16, 17, 18, 20], which alone is insufficient for FMs whose parameters exceed the capacity of a single device, or orient around alternative architectures, e.g., mixture of experts [19]. These alternative architectures provide promising directions for decentralized learning, however, they are currently only trained and evaluated on smaller datasets and at a smaller computational scale (e.g., MNIST and WikiText-2 in [19]) than their state-of-the-art counterparts, e.g., GLaM [21]. In this paper, we focus on a standard GPT-style architecture, without considering any changes that might alter the model architecture or the convergence behaviour during training.

To fulfill the potential of decentralization for the training of FMs, we need to be able to (1) take advantage of computational devices connected via heterogeneous networks with limited bandwidth and significant latency, and (2) support forms of parallelism beyond pure data parallelism. In this paper, we tackle one fundamental aspect of this goal — how can we assign different computational "tasklets", corresponding to a micro-batch and a subset of layers, to a collection of geo-distributed devices connected via heterogeneous, slow networks? This is not an easy task — even for fast and homogeneous data center networks, such assignments are still an open ongoing research challenge [22, 23, 24, 25, 26]. For the heterogeneous setting, it becomes even more challenging as the size of the search space increases dramatically. In the homogeneous setting, the homogeneity of the edges in the communication graph reduces the search space into many equivalent classes representing allocation strategies with the same communication costs, enabling efficient polynomial runtime algorithms [23, 24, 22, 25, 26]; however, in the heterogeneous setting, one has to consider potentially exponentially many more distinct allocation strategies — as we will see later, because of the heterogeneity of the communication matrix, even the sub-problem of finding the best pipeline parallelism strategy equates to a hard open loop travelling salesman problem [27].

In this paper, we focus on this challenging scheduling problem of decentralized training of FMs over slow, heterogeneous networks, and make the following contributions:

- We study the problem of allocating distributed training jobs over a group of decentralized GPU devices connected via a slow heterogeneous network. More specifically:
    - To capture the complex communication cost for training FMs, we propose a natural, but novel, formulation involving decomposing the *cost model* into two levels: the first level is a *balanced graph partitioning* problem corresponding to the communication cost of data

parallelism, whereas the second level is a joint *graph matching* and *traveling salesman* problem corresponding to the communication cost of pipeline parallelism.

- – We propose a novel *scheduling algorithm* to search for the optimal allocation strategy given our cost model. Developing a direct solution to this optimization problem is hard; thus, we propose an efficient evolutionary algorithm based on a collection of novel heuristics, going beyond the traditional heuristics used in standard graph partitioning methods [28].

- We carefully designed and implemented a collection of *system optimizations* to hide communication within the computation to further reduce the impact of slow connections.[2]

- We conduct extensive experiments that represent different scenarios of collaborative decentralized learning, simulated by using network measurements from different geographical regions of AWS. In the worldwide setting with 64 GPUs across 8 regions (Oregon, Virginia, Ohio, Tokyo, Seoul, London, Frankfurt, Ireland), we show that our system is 3.8-4.8× faster, in end-to-end runtime, than the state-of-the-art systems, for training GPT3-1.3B, *without any difference in what is computed or convergence dynamics*. In addition, we also provide careful ablation studies to show the individual effectiveness of the scheduler and system optimizations.

- We shed light on the potential of decentralized learning — our prototype in the global heterogeneous setting is only *1.7-3.5× slower than Megatron/Deepspeed in data centers even though its network can be 100× slower*. We hope this paper can inspire future explorations of decentralized learning for FMs, over geo-distributed servers, desktops, laptops, or even mobile devices.

**Limitations and Moving Forward.** In this paper, we tackle one foundational aspect of decentralized learning but leave as future work many problems that are important for a practical system. We assume that communication between devices is relatively stable for a reasonable amount of time and that all devices are always online without failure or eviction. Note that we also do not train a full system to full convergence, instead running partial training to confirm intermediate result equivalence across regimes. Scheduling over a dynamic, heterogeneous environment and providing fault tolerance, potentially with checkpointing, while training to convergence are directions for future exploration.

## 2 Decentralized Training of Foundation Models: Problem Formulation

We first introduce concepts, technical terms, and the procedure of decentralized training. Then we formally define the scheduling problem this paper tackles.

**Decentralized setting.** We assume a group of devices (GPUs) participating in collaborative training of a foundation model. Each pair of devices has a connection with potentially different delay and bandwidth. These devices can be geo-distributed, as illustrated in Figure 1, with vastly different pairwise communication bandwidth and latency. In decentralized training, all layers of a model are split into multiple *stages*, where each device handles a consecutive sequence of layers, e.g., several transformer blocks [29]. In addition, since the input for foundation model pre-training is huge, e.g., a few millions of tokens, it is also split into multiple *micro-batches* that can be handled in parallel.

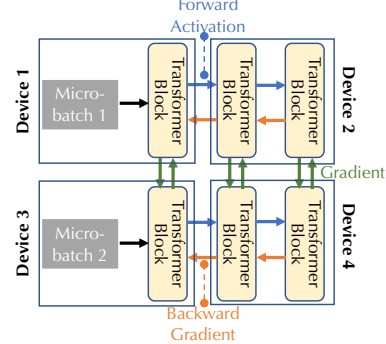

**Problem definition.** We define *tasklets* as a collection of computational tasks in foundation model training — Tasklet $t_{i,j}$ is the forward and backward computation for a stage $j$ with a micro-batch $i$ of training data in a training iteration. We aim to design an effective scheduler to assign each tasklet to a particular device so that the training throughput is maximized in decentralized training.

**Parallelism.** The above setting involves two forms of parallelism, *pipeline* and *data*. In pipeline parallelism, the compute in multiple stages is parallelized — each device handles activation or gradient computation for different micro-batches in parallel and the results can be communicated or passed to subsequent stages. Data parallelism means that devices compute the gradient for different micro-batches independently, but need to synchronize these gradients through communication. In a decentralized environment, the training procedure is *communication-bounded*. The scheduling problem is to accelerate the communication procedure by allocating tasklets that require high communication volumes between them to devices with faster connections.

---

[2]Our code is available at: `https://github.com/DS3Lab/DT-FM`.

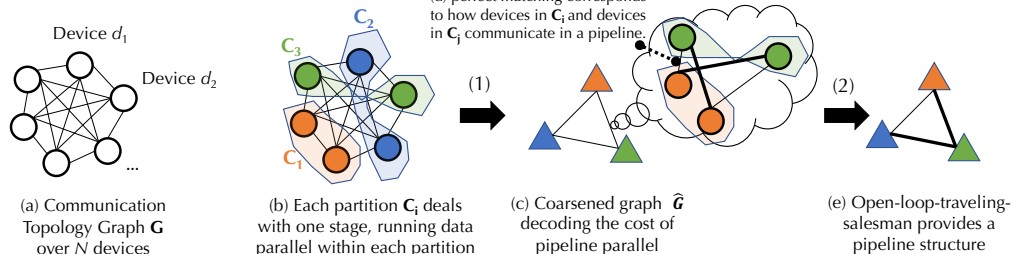

Figure 2: (a) Communication graph $\mathbf{G}$; and (b, c, d, e) an illustration of the cost model given $\mathbf{G}$.

**Formalization of the scheduling problem.** Formally, our scheduling problem is as follows.

- Let $\mathbf{D} = \{d_1 \ldots d_N\}$ be a set of $N$ devices; $\mathbf{A} \in \mathbb{R}_+^{N \times N}$ and $\mathbf{B} \in \mathbb{R}_+^{N \times N}$ be the communication matrix between these devices describing the delay and bandwidth respectively, where the delay and bandwidth between device $d$ and $d'$ is $\alpha_{d,d'}$ and $\beta_{d,d'}$.

- Given the communication matrix $\mathbf{A}$ and $\mathbf{B}$, we construct a communication graph $\mathbf{G}$ (Figure 2(a)) — each device corresponds to a node in $\mathbf{G}$ and each edge between $d$ and $d'$ is labeled with the average latency and bandwidth between $d$ and $d'$: $((\alpha_{d,d'} + \alpha_{d',d})/2, (\beta_{d,d'} + \beta_{d',d})/2)$. Even though $\mathbf{A}$ and $\mathbf{B}$ are asymmetric (i.e., upload and download speeds might be different), the communication graph $\mathbf{G}$ is symmetric because in our workloads all communications between two devices happen to involve the same amount of upload and download.

- The number of stages that a micro-batch needs to go through is $D_{\text{PP}}$ (noted as pipeline parallel degree); the number of batch partition that needs to run model gradient synchronization is $D_{\text{DP}}$ (noted as data parallel degree); we have $D_{\text{DP}} \times D_{\text{PP}} = N$, i.e., the total number of devices.

- $c_{\text{PP}}$ (resp. $c_{\text{DP}}$) represent the number of bytes of activations for a micro-batch (resp. parameters/gradients for a stage) communicated in pipeline parallelism (resp. data parallelism).

- We denote a training tasklet as $t_{i,j}$, where $i \in \{1, ..., D_{\text{DP}}\}$ and $j \in \{1, ..., D_{\text{PP}}\}$, each of which corresponds to one specific micro-batch $i$ and pipeline stage $j$.

- An assignment strategy $\sigma \in \mathbf{D}^{D_{\text{DP}} \times D_{\text{PP}}}$ assigns, for each tasklet $t_{i,j}$, a device $\sigma_{i,j} \in \mathbf{D}$, which means that device $\sigma_{i,j}$ runs the training tasklet $t_{i,j}$. A valid assignment needs to be unique, i.e., $\forall (i,j) \neq (i',j')$: $\sigma_{i,j} \neq \sigma_{i',j'}$. We use $\Sigma$ to denote the set of all valid assignments.

- An *optimal assignment strategy* is an assignment $\sigma$ that minimizes the communication cost

$$\sigma^* = \arg\min_{\sigma \in \Sigma} \text{COMM-COST}\,(\sigma)$$

**Challenges and Goals.** Our goal is to find the optimal assignment strategy, which involves two challenges: (1) How to effectively model the communication cost COMM-COST($\sigma$) for a given assignment $\sigma$ under a heterogeneous network environment? and (2) How to effectively search for the optimal assignment strategy that minimizes such a cost? We tackle these two questions in Section 3.

## 3 Scheduling in Heterogeneous Environments

Scheduling in the heterogeneous setting is a challenging task, as the size of the search space increases dramatically compared to that of the homogeneous case. In the homogeneous data-center case, the network delay can be usually ignored (e.g., $\mathbf{A} = \mathbf{0}$) and the bandwidth $\mathbf{B}$ are assumed to be formed by just a few constants — e.g., the communication bandwidths between different machines on the same rack are assumed to be same [23, 24, 22, 22, 26]. This significantly constrains the search space — one can ignore the influence of communication given uniform connections [23, 24, 22], or organize the device with a hierarchical structure [22, 26], making the scheduling problem solvable in polynomial time in terms of the number of machines.

In contrast, in the fully heterogeneous scenario the communication matrix $\mathbf{A}$ and $\mathbf{B}$ consists of distinct values, which can make the search space grows exponentially. In this section, we describe our scheduler that searches for an optimal strategy in the complex search space.

### 3.1 Overview of the scheduler

We carefully design a bi-level scheduling algorithm based on extended *balanced graph partition* problem (see Figure 2), and solve this problem by an evolutionary algorithm with a carefully designed

local search strategy. Given an assignment strategy $\sigma = \{\sigma_{i,j}\}$ for all tasklets $\{t_{i,j}\}$, we first model its communication cost. During the training of FMs, the communication costs come from two different sources: *(1) Data parallel*: All devices that are assigned with the tasklets dealing with the same stage $j$ (handling the same layers) of different micro-batches need to communicate within themselves to exchange gradients of these layers. For layer $j$, we call these devices its *data parallel group*: $\mathbf{C}_j = \{\sigma_{i,j} \mid \forall i \in [D_{\text{DP}}]\}$. We can implement the communication using different primitives, e.g., AllReduce [30], ScatterGatter [31], or other decentralized average protocols [16]. *(2) Pipeline parallel*: All devices that are assigned with the tasklets dealing with the same micro-batch $i$ of different stages need to form a pipeline, communicating within themselves to exchange activations and backward gradients. For micro-batch $i$, these devices are $\mathbf{P}_i = \{\sigma_{i,j} \mid \forall j \in [D_{\text{PP}}]\}$. Because these devices need to form a linear pipeline, any *permutation* over $\mathbf{P}_i$ corresponds to one strategy of how these machines can conduct pipeline parallelism within them.

**Scheduling Problem.** The goal of our scheduler is to minimize both costs. One design decision that we made is to decompose this complex optimization problem into two levels. At the first level, we consider the best way of forming $\mathbf{C}_j$'s, incurring data parallel communication costs within them. At the second level, we consider the cost of pipeline parallelism *given* an layout from the first level:

$$\min_{\mathbf{C}_1...\mathbf{C}_{D_{\text{PP}}}} \text{COMM-COST}\left(\mathbf{C}_1...\mathbf{C}_{D_{\text{PP}}}\right) := \text{DATAP-COST}(\mathbf{C}_1...\mathbf{C}_{D_{\text{PP}}})$$
$$+ \text{PIPELINEP-COST}(\mathbf{C}_1...\mathbf{C}_{D_{\text{PP}}}) \qquad (1)$$
$$s.t. \quad |\mathbf{C}_1| = .... = |\mathbf{C}_{D_{\text{PP}}}| = D_{\text{DP}}, \forall j, j' : \mathbf{C}_j \cap \mathbf{C}_{j'} = \emptyset, \mathbf{C}_1 \cup ... \cup \mathbf{C}_{D_{\text{PP}}} = \mathbf{D}$$

where computing PIPELINEP-COST($\mathbf{C}_1...\mathbf{C}_{D_{\text{PP}}}$) involves finding the optimal pipeline structure.

In Section 3.2 and Section 3.3, we provide details on COMM-COST $(\mathbf{C}_1...\mathbf{C}_{D_{\text{PP}}})$. Notice that this modified objective makes our problem different from the textbook graph partition problem; thus, we need a carefully designed evolutionary algorithm for finding such a solution introduced in Section 3.4.

### 3.2 Modelling data parallel communication cost

Given the communication graph $\mathbf{G}$ forming data parallel groups $\mathbf{C}_1...\mathbf{C}_{D_{\text{PP}}}$ corresponds to a *partition* of $\mathbf{G}$— In Figure 2(b), different colors correspond to devices in the same $\mathbf{C}_j$. The data parallel cost within $\mathbf{C}_j$ only relies on all communication channels (edges in the communication graph) connecting devices in $\mathbf{C}_j$. If we assume a colocated sharded parameter server [31] implementation for communicating within each $\mathbf{C}_j$, and recall that $c_{\text{DP}}$ represents the total amount of data (in bytes) that needs to be exchanged during gradient aggregation — each device in $\mathbf{C}_j$ needs to manage $c_{\text{DP}}/D_{\text{DP}}$ bytes of parameter shard. Once the gradient is ready, each device needs to send each of its local shards to the corresponding device; next, each device can aggregate the gradients it receives from all other devices in $\mathbf{C}_j$; and finally, each device will send the aggregated gradient shard to all other devices. Therefore, we can model the data parallel cost for $\mathbf{C}_j$ as follows:

$$\text{DATAP-COST}(\mathbf{C}_j) = \max_{d \in \mathbf{C}_j} \sum_{d' \in \mathbf{C}_j - \{d\}} 2 \cdot \left(\alpha_{d,d'} + \frac{c_{\text{dp}}}{D_{\text{DP}}\beta_{d,d'}}\right). \qquad (2)$$

Here, the total cost is bounded by the *slowest* device ($\max_{d \in \mathbf{C}_j}$), which needs to exchange data with all other machines ($\sum_{d' \in \mathbf{C}_j - \{d\}}$). Because the communication of these different data parallel groups $\mathbf{C}_1...\mathbf{C}_{D_{\text{PP}}}$ can be conducted in parallel and we are only bounded by the slowest data parallel group. This allows us to model the total communication cost for data parallelism as:

$$\text{DATAP-COST}(\mathbf{C}_1...\mathbf{C}_{D_{\text{PP}}}) = \max_{j \in [D_{\text{PP}}]} \text{DATAP-COST}(\mathbf{C}_j)$$

### 3.3 Modeling pipeline parallel communication cost

Given $\mathbf{C}_1...\mathbf{C}_{D_{\text{PP}}}$, to model the communication cost of pipeline parallelism, we need to consider two factors: (1) each *permutation* $\pi$ of $\{\mathbf{C}_1...\mathbf{C}_{D_{\text{PP}}}\}$ corresponds to a specific pipeline strategy — devices in $\mathbf{C}_{\pi_j}$ and devices in $\mathbf{C}_{\pi_{j+1}}$ communicates to exchange activations (during forward pass) and

gradients on activations (during backward pass); and (2) devices in $\mathbf{C}_{\pi_j}$ and devices in $\mathbf{C}_{\pi_{j+1}}$ need to be "matched" — only devices that are dealing with the same micro-batch needs to communicate. This makes modeling the cost of pipeline parallel communication more complex.

To model the cost of pipeline parallel communication, we first consider the best possible way that devices in $\mathbf{C}_j$ and $\mathbf{C}_{j'}$ can be matched. We do this by creating a *coarsened communication graph* (Figure 2(c)). A coarsened communication graph $\widehat{\mathbf{G}}_{\mathbf{C}_1...\mathbf{C}_{D_{\mathrm{PP}}}}$ is a fully connected graph, and each partition $\mathbf{C}_j$ in the original communication graph $\mathbf{G}$ corresponds to a node in $\widehat{\mathbf{G}}_{\mathbf{C}_1...\mathbf{C}_{D_{\mathrm{PP}}}}$.

In the coarsened graph $\widehat{\mathbf{G}}$, the weight on an edge between $\mathbf{C}_j$ and $\mathbf{C}_{j'}$ corresponds to the following — *if $\mathbf{C}_j$ and $\mathbf{C}_{j'}$ need to communicate in a pipeline, what is the communicate cost of the optimal matching strategy between devices in $\mathbf{C}_j$ and devices in $\mathbf{C}_{j'}$?* Recall that $c_{\mathrm{PP}}$ represents the amount of data between two devices for pipeline parallel communication, we can model this cost by

$$\min_{\mathcal{M}} \max_{(d,d') \in \mathcal{M}} 2 \left( \alpha_{d,d'} + \frac{c_{\mathrm{PP}}}{\beta_{d,d'}} \right) \tag{3}$$

where $\mathcal{M}$ is a perfect matching between $\mathbf{C}_j$ and $\mathbf{C}_{j'}$ — $(d,d') \in \mathcal{M}$ means that device $d \in \mathbf{C}_j$ will communicate with device $d' \in \mathbf{C}_{j'}$ (i.e., they deal with the same micro-batch). Computing this value is similar to the classical minimal sum weight perfect matching problem (MinSumWPM) in bipartite graphs [32], with the only difference being that we compute the *max* instead of the *sum*. As we will show in the supplementary material, similar to MinSumWPM, Eq 3 can also be solved in PTIME.

The coarsened communication graph captures the pipeline parallel communication cost between two groups of devices, *assuming* they become neighbors in the pipeline. Given this, we need to find an optimal permutation of $\mathbf{C}_1...\mathbf{C}_{D_{\mathrm{PP}}}$, corresponds to the structure of the pipeline. This becomes the *open-loop traveling salesman problem* [27] over this condensed graph (Figure 2(e)). Formally, we have the following definition of the pipeline parallel cost:

$$\mathrm{PIPELINEP\text{-}COST}\left(\mathbf{C}_1...\mathbf{C}_{D_{\mathrm{PP}}}\right) = \mathrm{OPENLOOPTSP}\left(\widehat{\mathbf{G}}_{\mathbf{C}_1...\mathbf{C}_{D_{\mathrm{PP}}}}\right) \tag{4}$$

where $\widehat{\mathbf{G}}_{\mathbf{C}_1...\mathbf{C}_{D_{\mathrm{PP}}}}$ is the coarsened graph defined above.

### 3.4 Searching via hybrid genetic algorithm

The scheduling problem solves the optimization problem in Eq 1, which corresponds to a *balanced graph partition* problem with a complex objective corresponding to the communication cost. Balanced graph partition problem is a challenging NP-hard problem [33]. Over the years, researchers have been tackling this problem via different ways [34, 35, 36]. We follow the line of research that uses hybrid genetic algorithm [37, 28] since it provides us the flexibility in dealing with complex objective.

**Hybrid Genetic Algorithm.** A hybrid genetic algorithm for balanced graph partition usually follows a structure as as follows. The input is a set of candidate balanced graph partitions which serves as the initial population. The algorithm generates the next generation as follows. It first generates a new "offspring" $o$ given two randomly selected "parents" $p_1$ and $p_2$. One popular way is to randomly swap some nodes between these two parents (we follow [28]). Given this offspring $o$, we then conduct local search starting at $o$ to find a new balanced partitioning strategy $o^*$ that leads to better cost. We then add $o^*$ to the population and remove the worst partition candidate in the population if $o^*$ has a better cost. As suggested by [37], the combination of heuristic-based local search algorithms and genetic algorithm can accelerate convergence by striking the balance between local and global optimum.

**Existing Local Search Strategy.** The key in designing this algorithm is to come up with a good local search strategy. For traditional graph partitioning task, one popular choice is to use the Kernighan-Lin Algorithm [38]. Which, at each iteration, tries to find a pair of nodes: $d$ in partition $\mathbf{C}_j$ and $d'$ in partition $\mathbf{C}_{j'}$, to swap. To find such a pair to swap, it uses the following "gain" function:

$$
\begin{aligned}
\mathrm{GAIN}_{KL}((d, \mathbf{C}_j) \leftrightarrow (d', \mathbf{C}_{j'})) = \sum_{d'' \in \mathbf{C}_{j'}} w_{d,d''} - \sum_{d'' \in \mathbf{C}_j - \{d\}} w_{d,d''} \\
+ \sum_{d'' \in \mathbf{C}_j} w_{d',d''} - \sum_{d'' \in \mathbf{C}_{j'} - \{d'\}} w_{d',d''} - 2w_{d,d'}
\end{aligned}
$$

where $w_{i,j}$ corresponds to the weight between node $i$ and $j$ in the graph. However, directly applying this local search strategy, as we will also show in the experiment (Section 4) does not work well. Greedily following $\text{GAIN}_{KL}$ does not decrease the communication cost of foundation model training. Therefore, we have to design a new local search strategy tailored to our cost model.

**Improving Local Search Strategy.** Our local search strategy is inspired by two observations:

1. Removing the device $d_1$ with a *fast* connection (say with $d_2$) within partition $\mathbf{C}_j$ will not tend to change the data parallel cost within $\mathbf{C}_j$, since it is only bounded by the slowest connections.
2. Once $d_1$ is moved to $\mathbf{C}_{j'}$, highly likely the pipeline parallel matching between $\mathbf{C}_j$ and $\mathbf{C}_{j'}$ will consist of the link $d_1 \leftrightarrow d_2$, since it is a fast connection.

Therefore, in our local search strategy we only consider the fastest connection within $\mathbf{C}_j$: $d_1 \leftrightarrow d_2$ and the fastest connection within $\mathbf{C}_{j'}$: $d'_1 \leftrightarrow d'_2$ and generate four swap candidates: $d_1 \leftrightarrow d'_1$, $d_1 \leftrightarrow d'_2$, $d_2 \leftrightarrow d'_1$, $d_2 \leftrightarrow d'_2$. We use the following gain function (take $d_1 \leftrightarrow d'_1$ as an example):

$$\text{GAIN}((d, \mathbf{C}_j) \leftrightarrow (d', \mathbf{C}_{j'})) = \frac{1}{|\mathbf{C}_{j'}|} \sum_{d'' \in \mathbf{C}_{j'}} w_{d_1, d''} - w_{d_1, d_2} + \frac{1}{|\mathbf{C}_j|} \sum_{d'' \in \mathbf{C}_j} w_{d'_1, d''} - w_{d'_1, d'_2}$$

where $\frac{1}{|\mathbf{C}_{j'}|} \sum_{d'' \in \mathbf{C}_{j'}} w_{d_1, d''}$ measures the *expected* pipeline parallel cost of connecting $d_1$ with other devices in $\mathbf{C}_{j'}$ *before the swap*, and $w_{d_1, d_2}$ is the cost of connecting $d_1$ with other devices in $\mathbf{C}_{j'}$ *after the swap*, assuming this fast link $d_1 \leftrightarrow d_2$ will now be used for pipeline parallelism.

Just like how Kernighan-Lin Algorithm [38] can be extended to a circular version [28] to swap multiple nodes beyond a pair, we can also extend our method into a circular one, following procedure as circular KL with our new gain function.

## 3.5 Other System Optimizations

We also have some system optimizations to further improve the performance. The most important optimization involves pipelining of communications and computations. We divide each stage in the pipeline into three slots: a receiving slot, a computation slot, and a sending slot. The receiving slot of stage $j$ needs to build connections to receive activations from the stage $j-1$ in forward propagation and to receive gradients of activations from stage $j+1$. The computation slot handles the computation in forward and backward propagation. Symmetric to the receiving slot, the sending slot of stage $j$ needs to build connections to send activations to stage $j+1$ in the forward propagation and send gradients of activations to stage $j-1$ in the backward propagations. These three slots are assigned to three CUDA streams so that they will be further pipelined efficiently; as a result, communication will overlap with computation. In the decentralized scenario (communication bound), computation can be fully hidden inside the communication time.

## 4 Evaluation

We demonstrate that our system can speed up foundation model training in decentralized setting. Specifically, (1) We show that our system is $4.8\times$ faster, in end-to-end runtime, than the state-of-the-art systems (Megatron and Deepspeed) training GPT3-1.3B in world-wide geo-distributed setting. Surprisingly, it is only $1.7 - 2.3\times$ slower than these systems in data centers. This indicates that we can bridge the gap between decentralized and data center training (up to $100\times$ slower networks) through scheduling and system optimization; (2) We demonstrate the necessity of our scheduler through an ablation study. We show that with the scheduler, our system is $2.7\times$ faster in world-wide geo-distributed setting.

## 4.1 Experimental Setup.

To simulate the decentralized setting, we use 8 different AWS regions (Oregon, Virginia, Ohio, Tokyo, Seoul, London, Frankfurt, and Ireland) and measure the latency and bandwidth between these regions (we consider the bandwidth that we can realistically obtain using NCCL and UDP hole punching between these regions). Given these measurements, we use 64 Tesla V100 GPUs and control their pairwise communication latency and bandwidth for five different cases:

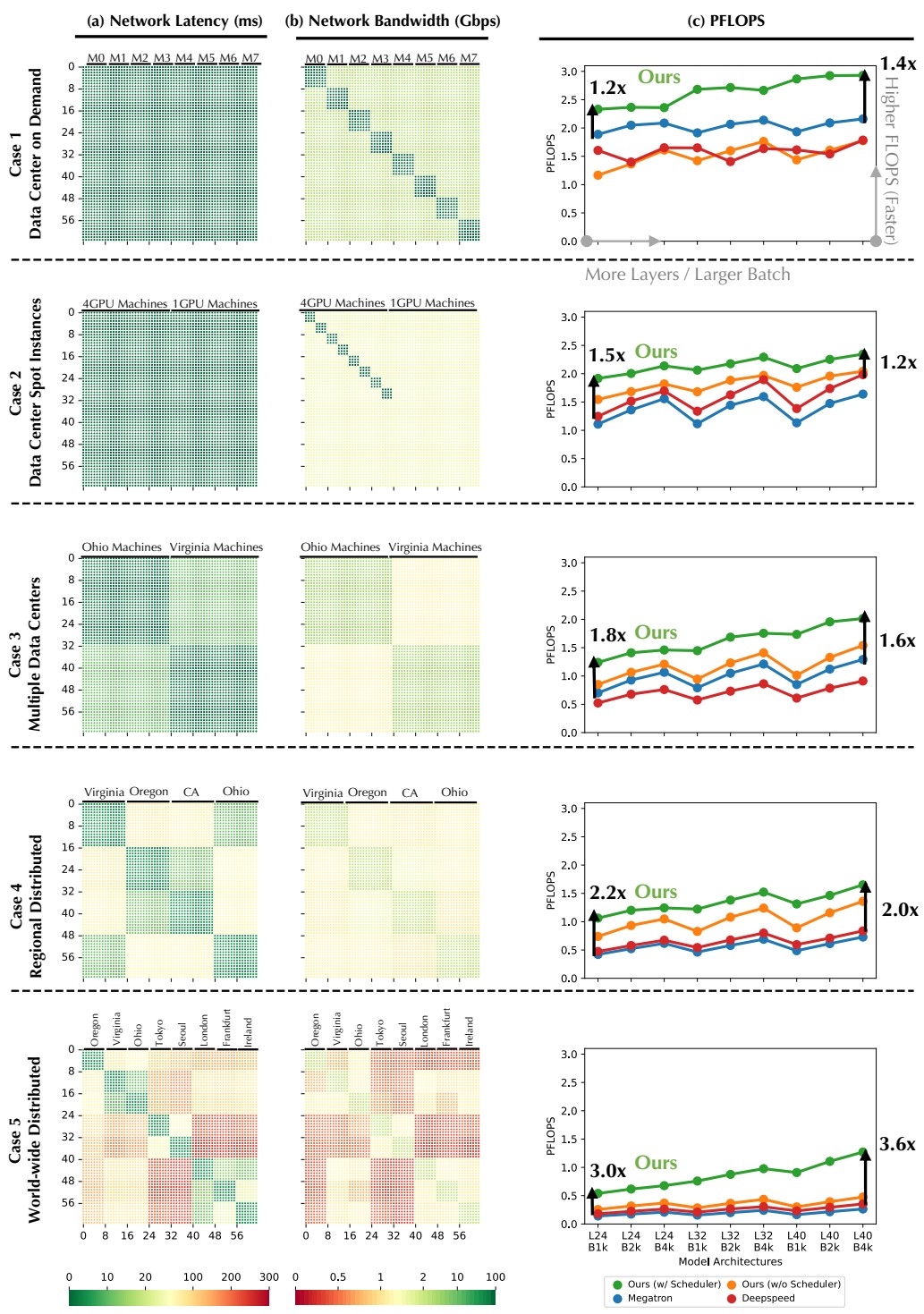

Figure 3: End to end compassion of our system with Megatron and Deepspeed in 5 different scenarios. Column (a) and (b) visualize the delay and bandwidth of 5 scenario respectively; Column (c) illustrate the comparison of Megatron, Deepspeed and our system w and w/o scheduler.

_Case 1. Data center on demand._ This is a standard setting that a user can obtain to train foundation models. we use 8 AWS `p3.16xlarge` nodes (each with 8 V100 GPUs); the intra-node connection is NVLink of 300 GB/s bi-directional bandwidth (150 GB/s unidirectional), and the inter-node connection has a bandwidth of 25 Gbps. We do not manually control latency and bandwidth here.

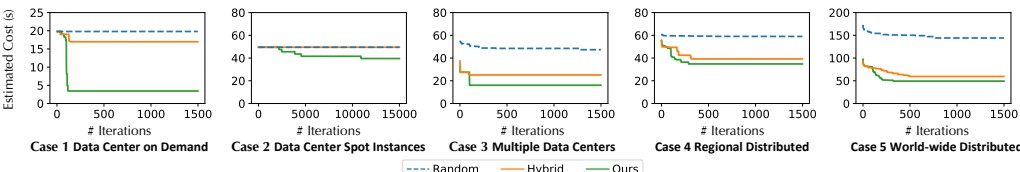

Figure 4: Comparison of Search Strategies.

*Case 2. Data center spot instances.* Spot GPUs are cheaper in a data center, but can be located on different types of machine. In this case, we rent 4 AWS `p3.8xlarge` nodes (each with 4 V100) and 32 `p3.2xlarge` nodes (each with 1 V100); the intra- `p3.8xlarge` node connection has a bandwidth of 100 Gbps, and the inter-node connection has a bandwidth of 10 Gbps. We do not manually control latency and bandwidth in this case.

*Case 3. Multiple Data Centers.* We consider two organizations, one in Ohio and another in Virginia, each organization contributes 32 V100 GPUs; within each organization, the bandwidth is 10 Gbps, and connections cross different campuses have a delay of 10 ms and bandwidth of 1.12 Gbps.

*Case 4. Regional geo-distributed.* We consider individual GPUs cross four different regions in US (California, Ohio, Oregon, and Virginia) ; within each region, the delay is 5 ms and bandwidth is 2 Gbps; cross different regions, the delay is 10∼70ms and the bandwidth is 1.0∼1.3 Gbps.

*Case 5. World-wide geo-distributed.* We consider individual GPUs cross eight different regions world-wide (Oregon, Virginia, Ohio, Tokyo, Seoul, London, Frankfurt, and Ireland); within each region, the delay is 5 ms and bandwidth is 2 Gbps; cross different regions, the delay is 10∼250ms and the bandwidth is 0.3∼1.3Gbps.

**Metrics and Model Architecture.** Since we do not introduce any optimizations that might change the computation or convergence, we can compare all methods by its throughput, we can compare all systems by the total number of floating point operations per second (PFLOPS), which is inverse proportional to the *runtime of each iteration* (which we show in Appendix). We use the standard GPT3-1.3B architecture [2], while also benchmarked different number of layers $\{24, 32, 40\}$, and batch sizes $\{1024, 2048, 4096\}$.

**Tuning of Megatron and Deepspeed.** We did a careful grid search of different parallelism settings and report the optimal results in each case—in Case 1, the optimal setting includes tensor model parallelism in Megatron and ZeRO-S3 in Deepspeed; in all other cases, the optimal settings are based on pipeline and data parallelism. We discuss more details in Appendix.

## 4.2 Results

**End-to-end Comparison.** Figure 3(c) shows the end-to-end comparison in terms of averaged PFLOPS achieved across different settings and different batch sizes and number of layers. In the world-wide geo-distributed cases, we achieve an $4.8\times$ speedup of Megatron, ($3.6\times$ speedup of Deepspeed). While in all other cases, our system can be $1.2 - 2.5\times$ faster. If we compare our system in Case 5 (world-wide geo-distributed) and Megatron/Deepspeed in Case 1 (data center on demand), it is exciting to see that the performance slowdown caused by decentralization is only $1.7 - 3.5\times$! This illustrates the great potential of decentralized training for foundation models. Additionally, Figure 3(c) illustrates another interesting behavior pattern. As increasing the batch size does not increases the communication cost of data parallelism and increasing # layers per device does not increases the communication cost of pipeline parallelism, with a larger batch size and a deeper model, the gap between centralized Megatron/Deepspeed and our decentralized system is even smaller.

**Effectiveness of Scheduler.** To evaluate the effectiveness of the scheduler, we disable it and use a random assignment in all cases and the results are also illustrated in Figure 3(c). We see that with our scheduler provides up to $2.7\times$ speeds up. To evaluate our local search strategy, we also compare our scheduler with a scheduler that uses the standard Kernighan-Lin algorithm for local search, illustrated in Figure 4. We see that, while both outperform random, our carefully designed local search strategy significantly outperforms Kernighan-Lin.

# 5 Related Work

**Foundation models.** Foundation models[1] refer to models that are trained on large-scale data and can be adapted (e.g., fine-tuned) to a wide range of downstream tasks. Current examples include BERT [39], GPT-3 [2], and CLIP[40]. Foundation models are usually trained in a data center, where the connection between GPUs is fast and homogeneous. ML infrastructures such as Megatron[4] and ZeRO[10, 11] have been proposed to distribute the training of these foundation models in a data center. Megatron uses `AllReduce` to synchronize activations in tensor model parallelism; ZeRO adopts `ScatterGather` to dispatch sharded parameters for layer-wise data parallelism. However, such collective communication paradigms would cause serious performance problems with slow and heterogeneous connections (see Appendix for detailed discussions).

**Decentralized optimization**. Decentralized training is first proposed within the scope of data parallelism, where each worker only synchronizes gradients with its neighbors (instead of all workers) to remove the latency bottleneck [17, 41, 16, 42, 43, 44]. Recently, [20] has also modified the implementation of data parallelism to support training in an open collaborative environment. Varuna [45] is released by Microsoft to support the training of GPT models in spot instances from a cloud service provider, which has the potential to be extended to the open collective scenario, but there is limited consideration with respect to the challenges of heterogeneous connections.

**Volunteer computing**. Distributing computationally intensive tasks over an open collaborative environment has been advocated for a few decades since the development of BOINC[46]; for example, the folding@home project [13] has been running simulations about protein dynamics on volunteers' personal computers for more than 20 years. Recently, the learning@home project[19] starts to consider training of mixture-of-expert transformers in such a volunteer computing setting.

# 6 Conclusion

In this paper, we probe the opportunity to train foundation models via a decentralized training regime with devices connected over a heterogeneous network. We propose an effective scheduling algorithm to assign tasklets from the foundation model pre-train computation. Empirical studies suggest that, in the worldwide geo-distributed scenario, our proposed scheduling algorithm enables a $4.8\times$ speed-up compared to prior state-of-the-art training systems. We believe that the decentralization and democratization of the training of FMs can shift the balance of power positively, but also necessitate new governance structures to help ensure the responsible development and deployment of FMs.

# Acknowledgments

CZ and the DS3Lab gratefully acknowledge the support from the Swiss State Secretariat for Education, Research and Innovation (SERI) under contract number MB22.00036 (for European Research Council (ERC) Starting Grant TRIDENT 101042665), the Swiss National Science Foundation (Project Number 200021_184628, and 197485), Innosuisse/SNF BRIDGE Discovery (Project Number 40B2-0_187132), European Union Horizon 2020 Research and Innovation Programme (DAPHNE, 957407), Botnar Research Centre for Child Health, Swiss Data Science Center, Alibaba, Cisco, eBay, Google Focused Research Awards, Kuaishou Inc., Oracle Labs, Zurich Insurance, and the Department of Computer Science at ETH Zurich. CR gratefully acknowledges the support of NIH under No. U54EB020405 (Mobilize), NSF under Nos. CCF1763315 (Beyond Sparsity), CCF1563078 (Volume to Velocity), and 1937301 (RTML); ARL under No. W911NF-21-2-0251 (Interactive Human-AI Teaming); ONR under No. N000141712266 (Unifying Weak Supervision); ONR N00014-20-1-2480: Understanding and Applying Non-Euclidean Geometry in Machine Learning; N000142012275 (NEPTUNE); NXP, Xilinx, LETI-CEA, Intel, IBM, Microsoft, NEC, Toshiba, TSMC, ARM, Hitachi, BASF, Accenture, Ericsson, Qualcomm, Analog Devices, Google Cloud, Salesforce, Total, the HAI-GCP Cloud Credits for Research program, the Stanford Data Science Initiative (SDSI), and members of the Stanford DAWN project: Facebook, Google, and VMWare. The U.S. Government is authorized to reproduce and distribute reprints for Governmental purposes notwithstanding any copyright notation thereon. Any opinions, findings, and conclusions or recommendations expressed in this material are those of the authors and do not necessarily reflect the views, policies, or endorsements, either expressed or implied, of NIH, ONR, or the U.S. This work was supported by an Open Philanthropy Award. The computation required in this work was provided by Together Computer https://together.xyz/.

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
