# A  Social Impact of Decentralized Training

In this paper, we find that decentralized training shows great potential for foundation models — such a technique would lead to significant positive social impacts. For example, decentralized training can utilize more inexpensive computational resources, which can significantly reduce the budget for the training of foundation models. This would increase the accessibility of foundation models for small research and commercial institutions. In fact, if the expense would be significantly reduced, large organizations would also receive some benefit from adopting such technology. On the other hand, we also notice that decentralization and democratization can also lead to a lack of control of cheaper computing resources and accelerate the risks of foundation models [1]. We look forward to actively engaging the community on governance questions.

# B  Limitation and Future Work

There are still some limitations of the current approach that could be explored further in future work.

First, we assume a homogeneous compute GPU resource in the scheduling algorithm, and in practice, different types of GPU could join the training computations. A native extension of the current solution could be to further split the tasklets into smaller pieces and to assign different numbers of pieces in different types of GPUs considering their memory budget and compute power as constraints. However, there are many opportunities for further improvement.

Second, there is still lots of room for the improvement of our scheduling algorithm, and strengthen our argument about the end-to-end speedup. In this paper, we hope to first provide some positive insights to the community that the potential of decentralized training can make it deployable for giant foundation models. On the other hand, there are many recent advances in system optimization that could lead to further improvement. For example, exploring recent advances in reinforcement learning [47] to solve our scheduling algorithm would be an interesting future direction. We believe that any improvement there can only improve the decentralized performance, which is consistent with the central message we try to share in this paper.

Last but not least, there are still some important open questions on the system side to handle the dynamics in decentralized environments. For example, some mechanism should be necessary to handle the dynamic join and leave of GPU nodes. On the other hand, failure happens more frequently in the decentralized environment, as fault tolerance should be considered for deployment, we believe that the current strategy such as checkpointing the model periodically could be adopted for this problem, but there would be some suitable solutions for the decentralized training runtime.

# C  Anatomy of the Current ML Systems for Foundation Model Training

Training foundation models [1] is a challenging task due to the enormous scale of such models — even the most powerful GPU cannot hold a complete copy of parameters for such models [48]. Thus, one cannot train such a model without distribution or using vanilla data parallelism.

Two popular approaches have been proposed to distribute the training of such foundation models in a data center:

- Megatron [4] distributes training by combining its proposed tensor model parallelism with pipeline parallelism [49, 22] and data parallelism [50]. The tensor model parallelism partitions individual layers across a group of workers and must run one `AllReduce` for the output activations of *each* layer in forward propagation and one `AllReduce` for the corresponding gradients in backward propagation for *each* layer.
- ZeRO [10] can be viewed as an effective optimization for data parallelism. The most effective mode is called ZeRO stage-3 from the Deepspeed implementation [5], and the equivalent implementation is known as Fully Sharded Data Parallelism (FSDP) from Fairscale [6]). In this mode, the parameter is sharded among all workers — in forward propagation, each worker conducts one `AllGather` to collect the parameters demanded for the current layer and discard the parameter after the forward computation; in backward propagation, each worker uses one `AllGather` to collect the parameter again and run one `ReduceScatter` to synchronize the gradients of this layer after the backward computation.

Both Megatron and ZeRO take a heavy usage of collective communication paradigms [51], which leads to two fundamental problems when it is deployed with *heterogeneous* and *slow* connections:

- *Demanding high bandwidth connections.* Both Megatron and ZeRO require high bandwidth connections for collective communications, since the compute cores are *idled* during communication slots. As long as communication takes an increasing share (due to lower bandwidth) of the execution time, the hardware efficiency drops dramatically. In fact, tensor model parallelism is recommended only within a single DGX server equipped with high-bandwidth NVLinks [48].
- *Sensitive to straggler.* The design and implementation of state-of-the-art collective communication libraries, e.g., NCCL [52], assume highly homogeneous connections within a data center, thus there is not sufficient robustness to handle the straggler among workers due to the heterogeneity of the open collective runtime. Furthermore, the layer-wise usage of collective communications in both Megatron and ZeRO has intensified this problem.

To bridge the performance gap between the data center and the decentralized open environment, we need to rethink the communication paradigms in different parallel strategies.

- *Pipeline parallelism is communication efficient.* Pipeline parallelism [49, 22, 53] partitions the model into multiple stages and a batch into multiple mini-batches, where once a worker finished the forward computation of a micro-batch, this worker will send the activations to the worker running the next stages; on the other hand, a worker needs to send the gradients of the activation back to the last stage in the backward propagation. Notice that pipeline parallelism utilizes *point-to-point* communications instead of collective paradigms. As long as one can put an increasing amount of computation inside a stage, the ratio of communication cost will also drop, leading to more efficient utilization of compute cores.[3] On the other hand, pipeline parallelism has its own limitation — one can only partition a model to a limited number of stages, which cannot scale out to lots of GPUs. We need to combine pipeline parallelism with data parallelism to scale out the training.
- *Scheduling is essential.* The *point-to-point* communication pattern in pipeline parallelism provides good opportunities to assign the training procedure on the decentralized environment that utilizes fast links and avoids slow links by a carefully designed scheduler, as presented in Section 3.

## D    Additional Details of Experimental Evaluation

We enumerate some additional details about our experiments.

### D.1    Multiple Execution of the Benchmark

We repeated all the benchmarks of 5 different scenarios listed in Section 4 three times. For our system with scheduler, since the scheduled layout is the same, we simply issued three independent executions in each scenarios; For our system without scheduler, we used three different random seeds (2022, 2023, 2024) to generate three layouts, and issued one execution for each layout in each scenario. The number in Figure 3 is based on an average of three different runs for each scenario — to avoid visual confusion, we did not plot the error bar within this line plot. We also repeated the scheduling algorithms three times with random seeds (0, 1, 2) to generate scheduled layouts and reported the average estimated cost (seconds) in Figure 4. In Figure 6, we plot *runtime of each iteration* as a bar chart with error bars. Notice that the variance of all executions in each setting is within $5\%$.

### D.2    Tuning of Megatron and Deepspeed

**Megatron.** We carefully tuned Megatron to perform a fair comparison with our system. As we mentioned in Section C. Megatron has three free degrees of parallel strategies: *tensor model parallelism*, *pipeline parallelism*, and *data parallelism*, we note the degrees of these parallel strategies as $D_{\text{TP}}$, $D_{\text{PP}}$, and $D_{\text{DP}}$ respectively. We conduct a complete grid search of the combinations of these hyper-parameters in the space of:

$$\{(D_{\text{TP}}, D_{\text{PP}}, D_{\text{DP}}) \,|\, D_{\text{TP}}, D_{\text{PP}}, D_{\text{DP}} \in \{1, 2, 4, 8\} \text{ and } D_{\text{TP}} \times D_{\text{PP}} \times D_{\text{DP}} = 64\}\,.$$

---

[3]Notice this is not always true since the device memory is limited. However, one can offload [11] (e.g., activations and parameters) to host memory to perform training on larger models with limited GPU device memory. Furthermore, the offloading through PCI-e is much faster compared to the decentralized connections, although it is slower than NVLink between GPUs in a data center.

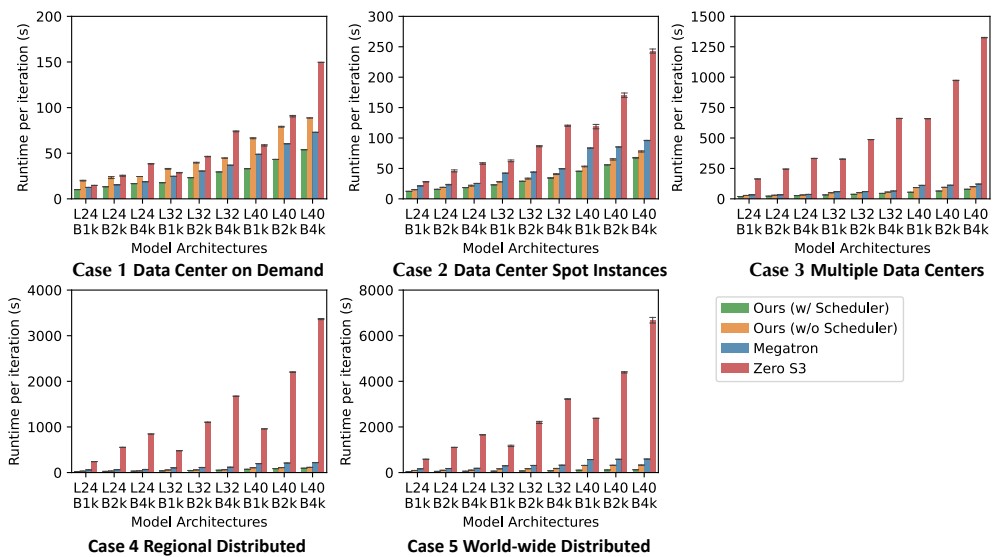

Figure 5: The performance of ZeRO-S3 from Deepspeed in terms of *runtime of each iteration* in 5 different Scenarios.

And we reported the optimal setting as the results for Megatron. Interestingly, only in Case 1 (data center on demand), the optimal setting includes tensor model parallelism (i.e., $D_{\mathrm{TP}} \neq 1$), where $D_{\mathrm{TP}} = 2, D_{\mathrm{PP}} = 4, D_{\mathrm{DP}} = 8$; in all other scenarios, the optimal setting is $D_{\mathrm{TP}} = 1, D_{\mathrm{PP}} = 8, D_{\mathrm{DP}} = 8$. This illustrates that tensor model parallelism is not suitable in slow and heterogeneous settings, consistent with our analysis in Section C. Since Megatron does not include a similar scheduler as its own components, we use the same random layouts as what we test for our system without scheduler.

**Deepspeed.** To run Deepspeed, we start with ZeRO-S3, which is usually viewed as the most significant technical contribution of the Deepspeed system. Under the same settings, the execution time of ZeRO-S3 is much longer comparing to both Megatron and our system (See Figure 5). This is consistent with our analysis in Appendix B. We then try to combine the pipeline parallel implementation in Deepspeed with its different implementations of data parallelism (e.g., ZeRO-S 1, 2 and 3)—it turns out that even the latest version of Deepspeed (0.6.7) only allows ZeRO-S1 to combine with pipeline parallelism. We find this combination outperforms ZeRO-S3 in most of the settings in Case 1 and all settings in Case 2,3,4 and 5. Notice that results of Deepspeed we report in Figure 3 is based on the optimal result of these two settings.

### D.3    Network Benchmark.

To obtain the network delay and bandwidth between different regions across the world, we rent AWS instances in 9 different data centers (California, Oregon, Virginia, Ohio, Tokyo, Seoul, London, Frankfurt, and Ireland). Instead of using AWS VPC, we setup our own VPN (using StrongSwan [54]) established on the public IP of these instances—any GPU machine connected to Internet can be linked in the same way. The strongSwan VPN would expose a private IP associated with a visible network interface, we can bind the NCCL communication on this network interface. The delay and bandwidth we obtained for cross-region NCCL connections are summarized in Table 1 for Case 4 regional geo-distributed scenario and and Table 2 for Case 5 world-wide geo-distributed scenario.

### D.4    Other Presentation of Experimental Results

In Figure 6, we plot *runtime of each iteration* for each scenario; this is a supplement to Figure 3.

### D.5    Deployment on FluidStack

We believe that having more realistic measurements and an end-to-end run can provide more pervasive statements for decentralized training. To this end, we conducted an additional experiment.

| Delay (ms) | | | | |
|---|---|---|---|---|
| | California | Ohio | Oregon | Virginia |
| California | - | 52 | 12 | 59 |
| Ohio | 52 | - | 49 | 11 |
| Oregon | 12 | 49 | - | 67 |
| Virginia | 59 | 11 | 67 | - |
| Bandwidth (Gbps) | | | | |
| | California | Ohio | Oregon | Virginia |
| California | - | 1.02 | 1.25 | 1.05 |
| Ohio | 1.02 | - | 1.10 | 1.12 |
| Oregon | 1.25 | 1.10 | - | 1.15 |
| Virginia | 1.05 | 1.12 | 1.15 | - |

Table 1: Delay (in milliseconds) and bandwidth (in Gbps) obtained by NCCL for Case 4 regional distributed scenario.

| Delay (ms) | | | | | | | | |
|---|---|---|---|---|---|---|---|---|
| | Oregon | Virginia | Ohio | Tokyo | Seoul | London | Frankfurt | Ireland |
| Oregon | - | 67 | 49 | 96 | 124 | 136 | 143 | 124 |
| Virginia | 67 | - | 11 | 143 | 172 | 76 | 90 | 67 |
| Ohio | 49 | 11 | - | 130 | 159 | 86 | 99 | 77 |
| Tokyo | 96 | 143 | 130 | - | 34 | 210 | 235 | 199 |
| Seoul | 124 | 172 | 159 | 34 | - | 238 | 235 | 228 |
| London | 136 | 76 | 86 | 210 | 238 | - | 14 | 12 |
| Frankfurt | 143 | 90 | 99 | 235 | 235 | 14 | - | 24 |
| Ireland | 124 | 67 | 77 | 199 | 228 | 12 | 24 | - |
| Bandwidth (Gbps) | | | | | | | | |
| | Oregon | Virginia | Ohio | Tokyo | Seoul | London | Frankfurt | Ireland |
| Oregon | - | 1.15 | 1.10 | 0.523 | 0.46 | 0.42 | 0.404 | 0.482 |
| Virginia | 1.15 | - | 1.12 | 0.524 | 0.500 | 0.364 | 1.02 | 1.05 |
| Ohio | 1.10 | 1.12 | - | 0.694 | 0.529 | 1.05 | 0.799 | 1.14 |
| Tokyo | 0.523 | 0.524 | 0.694 | - | 1.1 | 0.366 | 0.36 | 0.465 |
| Seoul | 0.46 | 0.500 | 0.529 | 1.1 | - | 0.342 | 0.358 | 0.335 |
| London | 0.42 | 0.364 | 1.05 | 0.366 | 0.342 | - | 1.14 | 1.09 |
| Frankfurt | 0.404 | 1.02 | 0.799 | 0.36 | 0.358 | 1.14 | - | 1.08 |
| Ireland | 0.482 | 1.05 | 1.14 | 0.465 | 0.335 | 1.09 | 1.08 | - |

Table 2: Delay (in milliseconds) and bandwidth (in Gbps) obtained by NCCL for Case 5 world-wide distributed scenario.

We rent 32 A40 GPUs (each with 48GB GPU memory, and 149.7 peak FP16 TFLOPS) from FluidStack [55], which consists of a group of geo-distributed GPU clusters, in (1) US Mid and (2) US East. We get the communication delay and bandwidths between GPUs as below:

- Intra-US Mid: delay $0.5_{\pm 0.1}$ ms; bandwidth $10.40_{\pm 1.11}$ Gbps;

- Intra-US East: delay $0.5_{\pm 0.1}$ ms; bandwidth $11.98_{\pm 1.92}$ Gbps;

- US Mid to US East: delay $21.8_{\pm 0.3}$ ms; bandwidth $3.87_{\pm 1.07}$ Gbps;

- US East to US Mid: delay $21.8_{\pm 0.3}$ ms; bandwidth $3.73_{\pm 1.38}$ Gbps.

We conducted an end-to-end run of the same training task of GPT3-1.3B without artificially controlling the bandwidth and latency. We also explore the training tasks of larger scale GPT3 models, including GPT3-6.7B, and GPT3-13B with a batch size of 1024. The performance in terms of the total number of floating point operations per second (PFLOPS) and runtime of each iteration are illustrated in Figure 7. This is a promising result of decentralized training — for GPT3-1.3B model with 40

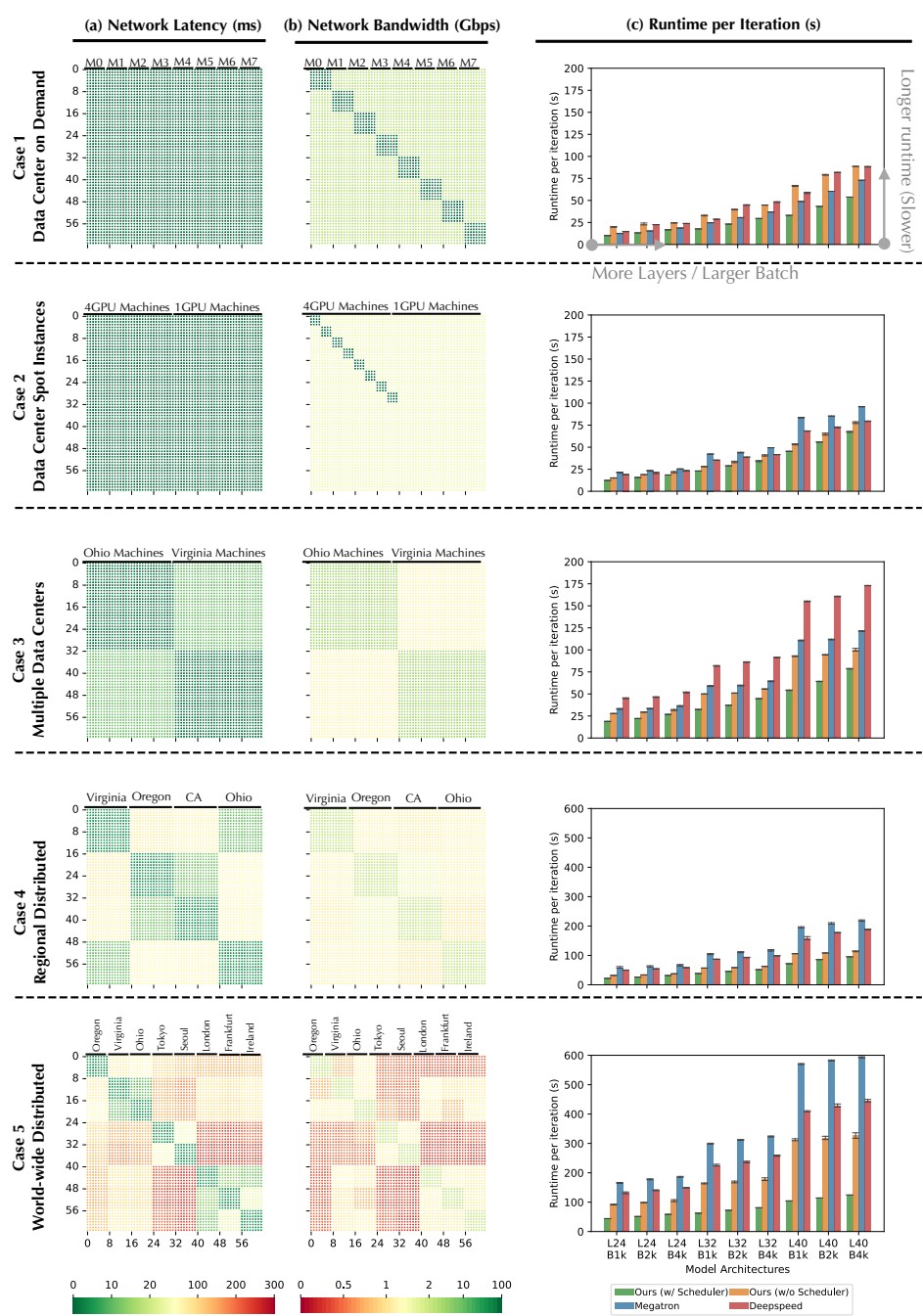

Figure 6: End to end compassion of in terms of *runtime of each iteration* in 5 different scenarios. We illustrate the comparison of Megatron, Deepspeed and our system with and without scheduler.

layers and 4K batch size, we archive $27.4\%$ of the peak FLOPS of the cluster, for GPT3-6.7B and GPT3-13B, we obtain $26.4\%$ and $29.7\%$ respectively.

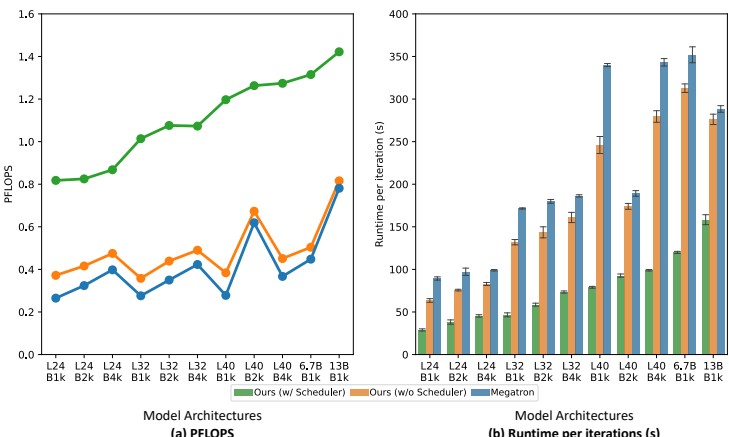

Figure 7: End to end compassion of in terms of the cluster's whole PFLOPS in (a) and runtime of each iteration in (b). We illustrate the comparison of Megatron and our system with and without scheduler.