# OpenReview forum: "Decentralized Training of Foundation Models in Heterogeneous Environments"
_NeurIPS.cc/2022/Conference — NeurIPS 2022 Accept_

### Official Review · Reviewer_TDE8 · 2022-07-11

**Rating:** 8
**Confidence:** 4
**Soundness:** 3 good
**Presentation:** 3 good
**Contribution:** 3 good

**Summary:**

Proposes techniques for scheduling training tasks in decentralized heterogenous clusters.

**Questions:**

* why not compare against other state of the art approaches such as

1. Jeff Rasley, Samyam Rajbhandari, Olatunji Ruwase, and Yuxiong He. Deepspeed: System optimizations enable training deep learning models with over 100 billion parameters. In Proceedings of the 26th ACM SIGKDD International Conference on Knowledge Discovery & Data Mining, pages 3505–3506, 2020

2. Deepak Narayanan, Mohammad Shoeybi, Jared Casper, Patrick LeGresley, Mostofa Patwary, Vijay Korthikanti, Dmitri Vainbrand, Prethvi Kashinkunti, Julie Bernauer, Bryan Catanzaro, et al. Efficient large-scale language model training on gpu clusters using megatron-lm. In Proceedings of the International Conference for High Performance Computing, Networking, Storage and Analysis, pages 1–15, 2021.

* It is not clear as to why there is a need for two-level optimization. Many of the cost functions are rather heuristic-driven. Is there a space for employing RL techniques for scheduling as suggested by others "Shyalika, Chathurangi, Thushari Silva, and Asoka Karunananda. "Reinforcement learning in dynamic task scheduling: A review."

**Limitations:**

* The related work and comparison could have been better.

* Exploration of newer techniques for optimization rather than using genetic algorithms

**Strengths And Weaknesses:**


Strengths
---
* achieves good speed up over Megatron in the decentralized setting
* only 2x slower in such decentralized setting even though the bandwidth/latency is several orders slower when going geo distributed
* 2 level scheduling models is intuitive

Weakness
---

* Ignores some of the latest innovation in Tensor model-parallel (TMP) strategies
* Ignores the fact that GPUs may itself be of different types across geos

---

> ### Author Response · Authors · 2022-08-02
> **To Reviewer TDE8**
>
> We thank the reviewer for the suggestions, which have helped us to improve the paper!
>
> **W1. “Ignores some of the latest innovation in Tensor model-parallel (TMP) strategies.”
> Q1. Why not compare against other state of the art approaches such as deepspeed and Megatron?
> L1. “The related work and comparison could have been better.”**
>
> **Response:** In this paper, we compared with Megatron as a strong baseline. We carefully tuned the combination of parallelism strategies in Megatron and pick the best one for each scenario, including (1) data parallel, (2) pipeline parallel, and (3) tensor model parallel. In data-center on demand setting, tensor model parallelism shows performance benefits, which is consistent with the insight from the Megatron paper, where TMP is only recommended within a single multi-GPU node, whereas other strategies should be adopted for inter-node parallelism.
>
> We definitely appreciate the suggestion to compare with DeepSpeed and believe it can make our paper stronger. We have added additional results of ZeRO-S3 from DeepSpeed (which is usually viewed as the most significant technical contribution of the deepspeed system) and included these results in Appendix D.5 (Figure 6). An example for Model-L40-B4K below:
>
> | 	   		       	   | Ours  (Runtime s) | Megatron | Zero S3 |
> |-----------------------------------------|----------------------|---------------|-------------|
> | Data Center on Demand 	  |	*53.8*	     |	72.96	 | 149.6    |
> | Data Center Spot Instances   |	*67.23*     |	96.16	 | 241.07  |
> | Multiple Data Centers 	         |	*78.79*     |	121.10	 | 1325.62 |
> | Regional Distributed 		  |	*95.15*     |	216.38 	 | 3377.46 |
> | Case 5 World-wide Distributed| 	*124.23*    |	595.64 	 | 6727.32 |
>
> In decentralized settings, we see that the execution time of ZeRO-S3 is much longer comparing to both Megatron and our system. This is consistent with our analysis in Appendix C, where we provide a detailed discussion about why TMP and fully sharded data parallelism (also noted as ZeRO-S3 from Deepspeed implementation) might not work well in a decentralized environment. In short, these approaches apply collective communication operator layer-wisely (allreduce in TMP; scatter-get and allgather in ZeRO-S3) — this is efficient under data center networks but can introduce significant overhead especially in slower networks.
>
> We are still running extensive configuration tuning for the Deepspeed system for different types of parallelism beyond ZeRO-S3 (e.g., ZeRO-S 1 and 2). In the decentralized setting, we expect deepspeed to not achieve significantly better performance than Megatron — it will highly likely resort to the same type of parallelism as Megatron under slow networks. We will report these results in a later revision.
>
> **W2. “Ignore the fact that GPUs may itself be of different types across geos”.**
>
> **Response:** We agree with the reviewer that our current scheduling algorithm does not consider this setting, but it is not a fundamental limitation. Actually we are very excited to address it in the future work to make our system more robust. We provide a more details of future work in Appendix B. Below is a partial discussion about system design.
>
> There are still some important open questions on the system side to handle the dynamics in decentralized environments. For example, some mechanism should be necessary to handle the dynamic join and leave of GPU nodes. On the other hand, failure happens more frequently in the decentralized environment, as fault tolerance should be considered for deployment, we believe that the current strategy such as checkpointing the model periodically could be adopted for this problem, but there would be some suitable solutions for the decentralized training runtime.
>
>
> **Q2. “It is not clear as to why there is a need for two-level optimization. Many of the cost functions are rather heuristic-driven. Is there a space for employing RL techniques for scheduling as suggested by others: Shyalika, Chathurangi, Thushari Silva, and Asoka Karunananda. "Reinforcement learning in dynamic task scheduling: A review."
> L2. “Exploration of newer techniques for optimization rather than using genetic algorithms.”**
>
> **Response:** Thanks for mentioning this direction! We agree that there can be lots of room for the improvement of our scheduling algorithm, and strengthen our argument about the end-to-end speedup. In this paper, we hope to first provide some positive insights to the community that the potential of decentralized training can make it deployable for foundation models.
>
> Exploring recent advances in RL to solve our scheduling algorithm would definitely be interesting — any improvement there can only make the decentralized performance better! We revised our paper to add this as a potential future direction in Appendix B. Partial discussion is provide below:

---

> > ### Comment · Reviewer_TDE8 · 2022-08-08
> > **Thank you for your response and additional experiments.**
> >
> > Thank you for your response and additional experiments.  I will update the score.

---

### Official Review · Reviewer_rKen · 2022-07-12

**Rating:** 8
**Confidence:** 5
**Soundness:** 4 excellent
**Presentation:** 4 excellent
**Contribution:** 4 excellent

**Summary:**

The work develops communication-efficient schedulers for parallel sub-units for the geo-distributed low-bandwidth high-latency training of GPT-2-style transformer language models. Both data-parallel sub-units (gradient communication) and pipeline-parallel sub-units (activation communication) are considered. The sub-units are optimized using statistics of latency and bandwidth together with a global genetic algorithm with local node-swap optimization (similar to Kernighan-Lin).


**Questions:**

Questions:

How does your approach scale vs Megatron as you scale the model?

Comments:
- Typo in line 226: generic = genetic
- Figure 4: It would be better to replace case 1,2... with On Demand, Spot, multiple DC, regional, world-wide. If you can find room, it would be beneficial to enlarge this figure as it is an important results that deserves a little more attention.
- It is better to refer to GPT3-XL by the number of parameters, that is, GPT-3 1.3B


**Limitations:**

A few limitations are discussed in the paper, but no separate limitation section exists. Ethical considerations should be explored. Making large model training more accessible to not require a supercomputer-quality network increases accessibility both for small institutions (democratization) but might also favor those that already have the most resources more (companies buying all the cloud GPUs that they can find world-wide, to train large neural networks cheaply)

**Strengths And Weaknesses:**

Strengths:

- The presentation is excellent. The writing is very clear, figures help with understanding and math is used appropriately.
- The evaluations are robust and extensive. The experiments were conducted in real geo-distributed settings.
- Ablations provide insight into baseline Kernighan-Lin search vs improved swapping algorithm

Weaknesses:
- Failure of nodes/GPUs is not discussed but mentioned as a limitation

This is one of the best papers that I read in this area for a while and I am happy to champion the paper and fight for its acceptance.

---

> ### Author Response · Authors · 2022-08-02
> **To Reviewer rKen**
>
> We thank the reviewer for the strong support! Our responses to your questions are as follows:
>
> **Q1. “How does your approach scale vs Megatron as you scale the model?”**
>
> **Response:** In Figure 4, we scale the workload in two ways: (1) more layers (L24 -> L40) and (2) larger batches (B1K -> B4K). As more compute load is added to the cluster, both our approach and Megatron get faster in terms of FLOPS. However, the increase of FLOPS in our approach is much more significant (Figure 4) given that the workload becomes less communication-bounded.
>
> **Q2-1. “Typo in line 226: generic = genetic”**
>
> **Response:** Thanks for noticing this; we have fixed it.
>
> **Q2-2. “Figure 4: It would be better to replace case 1,2... with On Demand, Spot, multiple DC, regional, world-wide. If you can find room, it would be beneficial to enlarge this figure as it is an important result that deserves a little more attention.”**
>
> **Response:** We have updated the manuscript with these notations to make the paper easier to follow.
>
> **Q2-3. “It is better to refer to GPT3-XL by the number of parameters, that is, GPT-3 1.3B.”**
>
> **Response:** Thanks for pointing this out; we have changed it to GPT3-1.3B.
>
> **W1. Failure of nodes/GPUs is not discussed but mentioned as a limitation.
> L1. “A few limitations are discussed in the paper, but no separate limitation section exists. Ethical considerations should be explored. Making large model training more accessible to not require a supercomputer-quality network increases accessibility both for small institutions (democratization) but might also favor those that already have the most resources more (companies buying all the cloud GPUs that they can find worldwide, to train large neural networks cheaply)”**
>
> **Response:** Thanks for mentioning these points! We added discussions of the social impact in Appendix A and a separate limitation section in Appendix B.
>
> Social impact: Decentralized training can utilize more inexpensive computational resources, which can significantly reduce the budget for the training of foundation models. This would increase the accessibility of foundation models for a small research and commercial institutions. However, we also notice that decentralization and democratization can also lead to a lack of control of cheaper compute resources and accelerate the risks of foundation models [1].
>
> There are still some limitations of the current approach that could be explored further in future work. For example, we assume a homogeneous compute GPU resource in the scheduling algorithm, and in practice, different types of GPU could join the training computations. Another example is that failure happens more frequently in the decentralized environment, so fault tolerance should be considered for deployment.
>
> [1] Bommasani et. al., On the Opportunities and Risks of Foundation Models.

---

### Official Review · Reviewer_BmFa · 2022-07-16

**Rating:** 7
**Confidence:** 4
**Soundness:** 4 excellent
**Presentation:** 3 good
**Contribution:** 3 good

**Summary:**

The authors propose a system for training large language models on hardware connected over the public internet and distributed around the world. This is a difficult problem (EleutherAI writes that “doing volunteer-driven distributed compute well for this use case is an unsolved problem”) and the authors tackle it with a combination of pipelining and data parallelism.

As this setting can’t assume homogeneous network performance, the parallelism assignment problem is much more difficult than it is in datacenters, and the authors apply a novel combination of algorithmic approaches to solve it, including treating pipelining and data parallelism independently and scheduling parallel units onto workers using an evolutionary algorithm.

The authors report benchmark results on a series of increasingly heterogeneous network configurations, showing significant speedup relative to a grid-search-optimized Megatron setup with homogeneous data and pipeline parallelism.


**Questions:**

How much do you lose by decomposing the assignment problem into separate data and pipeline levels? I think the answer might be “nothing”, but it would be great to be more explicit about that.

The intra-node bandwidth is quoted as 100 Gb/s in your experiments, but IIUC V100 NVlink is 150 GB/s (1200 Gb/s); is there a reason this isn’t the right number to use for intra-node bandwidth? And in general, are the bandwidths you’re assuming the result of benchmarks? If so, maybe include those benchmarks somewhere?

Were the benchmarks run on cloud machines that were really physically located in different regions, or were the bandwidths between machines limited artificially to simulate inter-region connections?


**Limitations:**

The authors are clear about most of the limitations of their system, including lack of online failure handling and dynamic replanning.

**Strengths And Weaknesses:**

The authors do a good job of motivating the problem and pointing out weaknesses in existing systems (e.g. a focus on data parallelism or mixture-of-experts models).

The paper uses the terminology “macro-batch” but it’s sometimes unclear whether this refers to the portion of the global batch that runs on one data-parallel replica, or to the portion that runs on one stage of one data-parallel replica (more often called a microbatch). I’d recommend being a little clearer here.

The algorithm the authors introduce seems reasonable and well-motivated. If it’s fast enough, and if decomposing the assignment problem into separate data and pipeline levels isn’t a restrictive assumption (both of these appear to be true!) then it also seems like the only algorithm we need for this problem. (It might be possible to optimize in an even greedier way, but if the evolutionary approach is fast enough I’m not sure it matters).

I’m somewhat disappointed that a system for heterogeneous distributed training (justified in part by a seemingly accurate claim that home electricity is 50-100x cheaper than spot GPU instances) was benchmarked only on cloud machines with cloud networking, not on machines with conventional home internet. But it looks like the bandwidth between distant datacenters in the authors’ experiments is low enough that it might be representative of home network connections.

The authors present good benchmarks against grid-search-optimized Megatron configurations, and their ablations, although brief, are convincing about the value of their local search strategy.

---

> ### Author Response · Authors · 2022-08-02
> **Additional response**
>
> **Q1. “How much do you lose by decomposing the assignment problem into separate data and pipeline levels? I think the answer might be “nothing”, but it would be great to be more explicit about that.”**
>
> **Response:** In theory, by decomposing the assignment into separate levels, our search space consists of a more restrictive family of schedules: it tends to favor schedules in which machines within the same stage finish with similar speed. In practice, our decompose seems to perform well, as illustrated by the experiments. To understand precisely how much we are losing with decomposition, we plan to conduct an exhaustive search on a smaller scale problem and compare it with our scheduler.
>
> **Q2. “The intra-node bandwidth is quoted as 100 Gb/s in your experiments, but IIUC V100 NVlink is 150 GB/s (1200 Gb/s); is there a reason this isn’t the right number to use for intra-node bandwidth? And in general, are the bandwidths you’re assuming the result of benchmarks? If so, maybe include those benchmarks somewhere?”**
>
> **Response:** Thanks for noticing this! This is a typo, and we apologize for it — it should have been 300 GB/s as the bi-directional bandwidth (150GB/s unidirectional): https://aws.amazon.com/blogs/compute/optimizing-deep-learning-on-p3-and-p3dn-with-efa/ We have fixed this in the paper.
>
> All network bandwidths that we are assuming are the result of real-world benchmarks: We establish a virtual network (using swam) among AWS machines in different data centers and measure the network bandwidth. We provide more details in Appendix D.3 (source code has also been included in the supplementary materials).
>
> We also conducted more measurements (see W2) during the author response period.
>
> **Q3. “Were the benchmarks run on cloud machines that were really physically located in different regions, or were the bandwidths between machines limited artificially to simulate inter-region connections?”**
>
> **Response:** The delay and bandwidth we adopted are measured on cloud machines that are physically located in different regions. We rent machines in different regions (California, Oregon, Virginia, Ohio, Tokyo, Seoul, London, Frankfurt, and Ireland) and establish a virtual network using swam. We then use linux traffic control to simulate the delay and bandwidth and set up an environment within the same region, and conduct our experiments.
>
> During the author response period, we also run on FluidStack with machines that are physically located in different regions and *without* limiting their inter-region connections. (See W2)

---

> ### Author Response · Authors · 2022-08-02
> **To Reviewer BmFa**
>
> We appreciate your great feedback! We have carefully thought through your questions and added corresponding experiments and detailed discussions in the updated paper. We provide details below:
>
> **W1. “The paper uses the terminology “macro-batch” but it’s sometimes unclear whether this refers to the portion of the global batch that runs on one data-parallel replica, or to the portion that runs on one stage of one data-parallel replica (more often called a microbatch). I’d recommend being a little clearer here.”**
>
> **Response:** We used macro-batch to refer to the portion of computation that runs on one stage of one data-parallel replica in the current version of the paper; we have switched to the term micro-batch, which is more widely adopted as the reviewer suggested.
>
> **W2. “I’m somewhat disappointed that a system for heterogeneous distributed training (justified in part by a seemingly accurate claim that home electricity is 50-100x cheaper than spot GPU instances) was benchmarked only on cloud machines with cloud networking, not on machines with conventional home internet. But it looks like the bandwidth between distant datacenters in the authors’ experiments is low enough that it might be representative of home network connections.”**
>
> **Response:** In this paper, we use cloud machines on AWS for both reproducibility and more controls in # GPUs and the communication matrix between these machines. The communication matrix that we are using are from measurements when these machines are geo-distributed and therefore, we believe that our experiments are representative.
>
> However, we do agree with the reviewer that having more realistic measurements and an end-to-end run can significantly strengthen this paper. To this end, we conducted two additional experiments.
>
> To illustrate that the assumed bandwidth in our paper reflects a realistic decentralized setting, we rented 6 machines (4-8 GPUs each) from Vast.ai, a platform on which users can rent out their own GPUs. We picked machines with relatively fast connections contributed by users from (1) St.-Petersbu, (2) Norway, (3) Vestland, and (4) Viken. We get communication bandwidths between 0.35Gbps - 0.67Gbps between these machines when they are not provided by the same user (otherwise it can be much faster). This is quite similar to what we are assuming in our paper, confirming the reviewer’s intuition that the bandwidth used in our experiments is low enough to be representative.
>
> Furthermore, we rented 32 GPUs from FluidStack, which consists of a group of geo-distributed GPU clusters, in (1) US West and (2) US East. We get communication bandwidths between 1.5Gbps - 4Gbps between these machines across different regions (up to 10Gbps within region). We conducted an end-to-en run without artificially controlling the bandwidth and latency. We observe similar performance as our Case 3 (Multiple Data Centers): ~0.9 PFLOPS with ~20% GPU utilitzation.
>
> We will add all these results to the Appendix.

---

> > ### Comment · Reviewer_BmFa · 2022-08-08
> > **Strong rebuttal**
> >
> > Thank you! The responses here and in your other comment go above and beyond to address my concerns. I'm bumping my rating up one slot.

---

### Author Response · Authors · 2022-08-02
**Revision Summary**

We thank all the reviewers for their insightful comments and suggestions, which will help improve this paper significantly. We are glad that reviewers found that the problem is well-motivated and the presentation is **excellent** [R1, R2]; our approach is **intuitive and achieves good performance** [R1, R2, R3]; evaluations are **extensive and insightful** [R1, R2, R3]; our paper is one of the **best** papers that they read in this area [R2]. We have updated the paper to incorporate this feedback. The major changes are as follows:

1. [R1] We conduct more measurements on real-world decentralized networks using vast.ai, a platform on which users can rent out their GPUs, to justify the bandwidth settings in our experiments. We also include more details about measuring the delay and bandwidth between different regions of AWS.


2. [R1] To address the concern that our experiments largely rely on artificially controlled bandwidth, we added additional experiments with a real-world decentralized run using FluidStack (without artificially controlling communication bandwidth and latency).


3. [R3] We added additional experiments comparing with DeepSpeed, specifically ZeRO-S3.


4. [R1] We analyze the potential loss of the decomposition in our algorithm.


5. [R2, R3] We add more concrete discussions about the limitation, social impact, and future work.

We look forward to the discussion phase with all the reviewers. We appreciate all reviewers’ comments and feedback!

---

### Meta-Review · Area_Chair_ppDg · 2022-08-21

**Recommendation:** Accept
**Confidence:** Certain

**Metareview:**

All of the reviewers felt that this is a strong submission.  The paper gives a new novel approach for scheduling decentralized training tasks.  This will be of general interest to the community.

**Award:**

Yes

---

### Decision · Program_Chairs · 2022-09-14

Accept